# Osteopathic Manipulative Medicine: A Brief Review of the Hands-On Treatment Approaches and Their Therapeutic Uses

**DOI:** 10.3390/medicines9050033

**Published:** 2022-04-27

**Authors:** Ashley Roberts, Kaylee Harris, Bethany Outen, Amar Bukvic, Ben Smith, Adam Schultz, Stephen Bergman, Debasis Mondal

**Affiliations:** 1DeBusk College of Osteopathic Medicine, Lincoln Memorial University, 9737 Cogdill Road, Knoxville, TN 37932, USA; ashley.roberts02@lmunet.edu (A.R.); kaylee.harris@lmunet.edu (K.H.); bethany.outen@lmunet.edu (B.O.); amar.bukvic@lmunet.edu (A.B.); ben.smith@lmunet.edu (B.S.); adam.schultz@lmunet.edu (A.S.); 2DeBusk College of Osteopathic Medicine, Lincoln Memorial University, 6965 Cumberland Gap Parkway, Harrogate, TN 37752, USA; stephen.bergman@lmunet.edu

**Keywords:** osteopathy, muscle energy, myofascial release, balanced ligamentous tension, diaphragm, HVLA (High-velocity low amplitude), lymphatic pump, cranial osteopathy

## Abstract

Osteopathic manipulative medicine (OMM) is an emerging practice in the healthcare field with increasing popularity and evidence-based therapy. Osteopathic manipulative treatments (OMT) include hands-on manipulations of different body structures to increase systemic homeostasis and total patient well-being. Indeed, this new realm of the whole patient-based approach is being taught in osteopathic schools around the country, and the osteopathic principles of a mind-body-spirit-based treatment are being instilled in many new Doctor of Osteopathy (D.O.) students. However, despite their proven therapeutic value, there are still many individuals, both in and outside the medical profession, who are unaware (or misinformed) of the therapeutic uses and potential benefits of OMT. Here, we provide a brief introduction to this osteopathic therapeutic approach, focusing on the hands-on techniques that are regularly implemented in the clinical setting. It is becoming increasingly evident that different OMTs can be implemented to enhance patient recovery, both alone and in conjunction with the targeted therapies used in allopathic regimens. Therefore, it may be beneficial to inform the general medical community and educate the public and those associated with the healthcare field about the benefits of using OMT as a treatment modality. OMT is lower-cost, noninvasive, and highly effective in promoting full-body healing by targeting the nervous, lymphatic, immune, and vascular systems. There is a growing body of literature related to osteopathic research and the possible molecular pathways involved in the healing process, and this burgeoning field of medicine is expected to increase in value in the healthcare field. This brief review article explains the frequently utilized OMT modalities and their recognized therapeutic benefits, which underscore the need to understand the possible molecular mechanisms and circulating biomarkers linked to the systemic benefits of osteopathic medicine.

## 1. A Brief Overview of Osteopathic Medicine

In principle, it is generally believed that osteopathy is an alternative practice of medicine that accentuates the body’s innate principles of self-healing and modification to achieve homeostasis [1]. Osteopathic therapy utilizes manipulation of the body’s tissues and bones to facilitate the healing process. The lower cost of osteopathic medicine, their noninvasive methodologies, and their proven efficacy in promoting full-body healing is a testament to the growing number of osteopathic medical schools around the country. There are currently 37 accredited colleges of osteopathic medicine in the United States. An estimated 7445 osteopathic medical students graduated in 2021, and 8945 first-year osteopathic medical students are expected to matriculate in the 2021-22 academic year [2]. One in four medical students attends a college of osteopathic medicine [2]. Upon graduating, they will join the record-breaking 121,000 + other D.O.s within the United States. Indeed, for the first time in history, osteopathic medical students make up more than 25 percent of the U.S. medical student population. Unfortunately, however, for those pursuing a career in osteopathic medicine, it has become increasingly evident that the lack of knowledge in this area of medical study, both within the general medical community and in the well-informed public, needs to be more thoroughly addressed, appreciated and disseminated. 

Often when introduced as a Doctor of Osteopathic Medicine, a frequent response by the general public is “what’s that?”. In this article, in addition to different osteopathic treatments, emphasis is placed on the motto of osteopathic medicine, i.e., the compassionate care bestowed by the Doctor of Osteopathy (D.O.). Previously, osteopaths practiced in their own clinics and hospitals, and while they still do, nowadays, osteopathic medicine is being incorporated into most healthcare facilities. The use of many osteopathic techniques is also becoming more common in certain areas of healthcare, especially primary care [3]. Therefore, the goal of this review article is to educate the public on osteopathic medicine, what it is and how it is used. 

Osteopathic tenets describe the body as a single, functioning unit. The goal of osteopathic medicine is to facilitate and support self-healing, which is encompassed by the four osteopathic tenets of medicine [1]. These four tenets of osteopathy are as follows: (i). The body is a single unit that includes the body, mind, and spirit; (ii). The body is capable of self-regulation, self-healing, and health maintenance; (iii). Structure and function are interrelated; and (iv). Rational treatment is based on the understanding and application of principles one through three. Much of the osteopathic approach entails listening to both the patient and the patient’s body. “Listening” to the patient’s body requires the knowledge of somatic dysfunction and the skill of physical diagnosis, which is the additional education offered in osteopathic schools. In this respect, osteopathic training creates a physician with increased dynamic active listening skills at the onset of medical practice. While osteopaths are trained similarly to allopathic physicians, there are additional learning components known as osteopathic manipulative treatments (OMTs) and the use of compassion and care to treat the patient as a whole and not just the diseased organ. There are many techniques in osteopathic medicine that address specific dysfunctions and maladies. 

### Osteopathy: Dysfunctions and Treatments

Utilizing the osteopathic tenets mentioned previously, the body as a unit includes the autonomic nervous system, which can be affected by somatic dysfunction. Although both osteopathic (DO) and allopathic (MD) physicians participate in the same medical school courses and residency programs and can specialize within any field of medicine, only DOs are uniquely trained in administering OMTs [3,4], which are a set of manual treatments that improve physiologic function and support homeostasis within the skeletal, arthrodial, and myofascial structures of the body [5].

An osteopathic physician indicates the use of specific sets of OMTs based on the diagnosis of somatic dysfunctions, such as tenderness, asymmetry, restriction in motion, or tissue-texture changes [3,5]. These somatic dysfunctions can occur due to impaired functions of the skeletal, arthrodial, and myofascial structures and their related vascular, lymphatic, and neural elements [1,5]. Dysfunctions can also occur within the spinal vertebrae and the autonomic nervous system and can initiate anywhere in the body, causing chronic pain and increased morbidity [6]. The mechanisms underlying the beneficial actions of different OMTs primarily rely on the interconnectedness of the body systems and the inherent capacity for self-healing. An abnormality with the musculoskeletal system can manifest in the organs (somato-visceral reflex), and pathology of the visceral organs can manifest as either reduced range of motion or tissue texture changes in the musculoskeletal system (viscero-somatic reflex) [6]. Administration of OMTs not only addresses these dysfunctions but also reduces pain, increases range of motion, increases the ability to move with ease, and improves neurovascular and lymphatic flow to facilitate the resultant benefits.

The osteopathic approach to patient care was intended to be a ‘complete system’. This means that the OMT prescription can be directed and modified for any disease type and severity. In this light, OMT is typically considered for utilization as an adjunct to conventional allopathic medical interventions. The adaptability of OMT lends it to the treatment of multiple disease types and severities. Osteopathic techniques are adjusted based on the patient’s condition, age, weight, and other characteristics, yielding personalized, holistic medicine. Techniques include manipulation of the lymphatics, rib raising, diaphragmatic manipulations, high-velocity low-amplitude, myofascial release, balanced ligamentous tension, muscle energy, and cranial concept, which is a unique and pivotal technique modality that belongs to osteopathic medicine (Table 1). There are additional concepts such as articulatory techniques (range of motion) and variations of activating forces such as springing or oscillating. However, this article covers a scope of technique and application that is applicable in a general setting.

## 2. Muscle Energy Techniques

### 2.1. Introduction

Osteopathic medicine encompasses many treatment modalities to target the diagnoses. A single diagnosis can be treated in a variety of ways, depending on many factors, such as the patient’s age, the severity of the diagnosis, the size of the patient and/or the physician, the setting, etc. A popular treatment modality of choice is muscle energy technique. This approach is both *direct* and *active*, where the patient actively participates in the treatment and is placed into his/her barrier or restrictive motion [7]. Muscle energy technique was invented in 1948 by Fred Mitchell Sr., D.O. [8]. This treatment is used primarily to increase the range of motion of restricted areas, stretch tight muscles, reduce pain, and improve circulation and lymphatic flow throughout the body [7,8]. Contraindications to muscle energy may include if the patient has had a recent surgery, has compromised strength, has a fracture or open wound on the area requiring treatment, or has chronic joint disease [8]. 

Muscle energy can be divided into two subtypes: post-isometric relaxation and reciprocal inhibition [9] (Figure 1). Isometric contractions are created as a muscle produces tension that does not change the length of a muscle (i.e., holding a weight in a constant position) [7,8]. Isotonic contractions are produced when a muscle changes in length and can be classified as concentric (shortening) or eccentric (lengthening) [7]. Post-isometric relaxation focuses on relaxing and lengthening a hypertonic muscle using the agonist muscle, while reciprocal inhibition accomplishes the same goal using the antagonist muscle. Both treatment setups begin the same: identify a somatic dysfunction. Consider the scenario of doing bicep curls at the gym; when the arm curls up toward the chest, it is producing a concentric, isotonic contraction, whereas when the arm is being lowered back down to the start point, it is participating in an eccentric, isotonic contraction.

The most important part of this muscle energy treatment method is to localize the area of greatest restriction and find the barrier [7]. To do so, the osteopathic physician will compare each joint/muscle group bilaterally and determine which appears restricted, asymmetric, tender, and/or has a difference in tissue texture (such as hot and boggy). For example, the patient may be complaining of difficulty moving his/her leg into abduction (away from the midline) (Figure 1). After finding the area of greatest restriction, the physician must determine the patient’s restrictive barrier [8]. To understand the mechanisms behind the muscle energy technique, one must first understand the varying types of muscle contractions. In this respect, muscle energy techniques can be divided into two subtypes, post-isometric relaxation and reciprocal inhibition, as explained below.

### 2.2. Post-Isometric Relaxation

For post-isometric relaxation, the physician will position the muscle into the barrier (the point where tension is first felt) and instruct the patient to contract his/her muscle in the direction of ease (away from the barrier) [8]. Consider again the example of limited abduction in the leg; the physician will direct the patient’s limb into abduction until tension is felt. Next, the physician will instruct the patient to adduct his/her leg into the midline. The contraction will be held against the physician’s hand for approximately five seconds. The physician then instructs the patient to relax and moves the limb farther into abduction until tension is felt once again. The patient will contract again for five seconds against counterforce produced by the physician. The process is repeated three to five times or until the desired range of motion is achieved [8]. 

### 2.3. Reciprocal Inhibition

While the goal of reciprocal inhibition is the same as post-isometric relaxation, the mechanism is different. Using this method, the physician would instruct the patient to contract the antagonist muscle to trigger relaxation and stretch of the agonist muscle [8]. Consider a hypertonic bicep. The physician will place the patient into the restrictive barrier just as one would for post-isometric relaxation, but the physician will have the patient contract the triceps muscle against his/her force rather than contracting the bicep muscle (as would be done during post-isometric relaxation) [8]. The physician will then place the patient into the new restrictive barrier and continue the treatment as previously described. The difference between the subtypes is which muscle is being used to yield results—the agonist or the antagonist [9].

The underlying mechanism of post-isometric relaxation involves the Golgi tendon organs of the muscle being treated [9]. When the muscle is being contracted during the treatment, the Golgi tendon organ is stimulated. The stimulation sends a signal through the 1b afferent neurons to the inhibitory interneurons in the spinal cord. This signal creates a refractory period following muscle contraction and initiates a reflexive relaxation on the target muscle, which is why the physician can position the patient into a new barrier [8]. The reciprocal inhibition modality does not rely on the Golgi tendon organ as post-isometric relaxation does; rather, it activates muscle spindle fibers through stretch [8,9,10]. The stretch stimulates the muscle spindle, which in turn activates 1a afferent motor neurons [10]. The key difference between post-isometric relaxation and reciprocal inhibition is that the former is the ability of a muscle to relax when it experiences a stretch or increased tension, while reciprocal inhibition is the relaxation of muscles on one side of a joint to accommodate contraction on the other side of that joint. The result is an inhibition of the alpha motor neurons sending information to the agonist muscle, leading to relaxation of the muscle. Therefore, the physician can indirectly treat the hypertonic muscle by the contraction of the antagonist muscle, leading to relaxation of the agonist muscle (Figure 2). 

### 2.4. Diagnosing Somatic Dysfunctions

To begin muscle energy of the spine, the physician must diagnose a somatic dysfunction. The diagnosis is made hands-on by localizing the area of greatest restriction and identifying the vertebral segment or group of vertebrae causing the restriction [11]. Once the area of greatest restriction has been identified, a localized diagnosis of positional preference in 3 dimensions/planes is the next step. This will help to determine the technique setup by identifying the position of the barrier. First, the physician will place his/her thumbs on each transverse process of the vertebra and determine which transverse process is more posterior compared to the other. The posterior transverse process determines the direction of rotation of the vertebral segment. If the right transverse process is more posterior, the vertebra is rotated to the right. Next, the physician will assess the symmetry of the transverse processes in a neutral, flexed, and extended position. Whichever position provides the most symmetry between the processes is deemed the diagnosis. For example, if the symmetry improves when the patient flexes his/her back, the patient has a flexion dysfunction—named for its direction of ease [11]. If the symmetry improves with flexion or extension, the side bending component of the vertebra will be the same as the rotation component, and the dysfunction is likely localized to a single segment. If the symmetry remains unchanged, the side bending component is the opposite of the rotation component, and the diagnosis is a neutral dysfunction, which is more likely to present as a group dysfunction [12].

Once a somatic dysfunction has been diagnosed, the physician can then set the patient up for OMT. If using the post-isometric muscle energy treatment, the patient will be placed into his/her restrictive barrier and will then apply contraction in the direction of ease against a counterforce applied by the physician for approximately five seconds, relax, and repeat (Figure 3). In the case of the vertebral column, the restrictive barrier will have three components: flexion/extension/neutral, rotation, and side bending [11] (Figure 3). 

### 2.5. Uses for Muscle Energy Technique

Muscle energy technique can be used to treat many dysfunctions. One of the most prevalent presentations it targets is lower back pain [13]. Low back pain has been identified as the leading cause of disability in the U.S. Treating this pain with muscle energy limits the prescription of medications, specifically opioids, as well as invasive (and oftentimes unnecessary) surgeries and expensive imaging, and it allows the patient to sustain a better quality of life. Indeed, a number of studies have shown chronic pain to correlate with more than just physical symptoms, but mental symptoms as well, such as anxiety and depression disorders [11]. Hence, the use of OMT as a treatment modality allows easier access and more affordable care options.

### 2.6. Clinical Trials Studying Muscle Energy Technique

Due to the prevalence of low back pain in the U.S., the effectiveness of muscle energy technique has been investigated in a series of trials to deem it a fit course of treatment [11,12]. Low back pain can be experienced due to dysfunctions of the vertebral column or a number of other dysfunctions, such as those of the ribs, sacrum, pelvis, and lower extremities. Fortunately, muscle energy can be utilized to treat all of these dysfunctions. There have been numerous case studies conducted to determine its effectiveness. A pilot clinical trial investigated the impact of muscle energy to treat low back pain in a population of 19 participants, equally divided into a control and experimental group [13]. At the conclusion of the study, the data showed a statistically significant difference between groups, with the treatment group showing a larger decrease in pain. A systemic review also demonstrated the success of muscle energy treatment on acute and chronic low back pain as well as other musculoskeletal dysfunctions [14]. In 2016, a randomized controlled trial deemed muscle energy technique effective in treating nonspecific low back pain before advancing to more expensive, invasive relief methods [15]. A recent study in 2018 compared the effectiveness of treating patients with muscle energy technique compared to strain-counter strain technique and found both to be successful treatment modalities to alleviate low back pain [16]. Another case study recorded the treatment progress of a patient experiencing low back pain; the patient was treated using muscle energy technique on the sacroiliac joint rather than on the lumbar spine itself, and the study reported realignment of the patient’s pelvis after a series of treatments and, consequently, reduced low back pain [17]. The latter case study clearly demonstrated one of the underlying principles of osteopathic medicine: addressing the body as a single, connected unit. 

Treatment of the deficit caused by dysfunctions in the sacroiliac region is critical for rapid recovery from pain and long-term patient well-being. Consider the sacroiliac joint—the anatomical structure responsible for holding the body upright and movement of the lower extremities. The method of treating low back pain with sacroiliac muscle energy techniques is demonstrated in a study comparing the OMT to conventional therapies, and the study supported the use of muscle energy technique over conventional therapies [18]. A study by Patel et al. (2018) showed that the muscle energy technique can be used to correct the rotation of the innominate (pelvic bone). Upon restoration of the innominate, the sacroiliac joint was relieved of tension, and the low back pain was also relieved as a result [18]. 

Muscle energy technique can also be used to treat visceral conditions/diseases. In an eloquent study in 2019, the effectiveness of muscle energy technique in treating chronic obstructive pulmonary disease (COPD) was assessed [19]. The results of the study did not indicate muscle energy treatment as an effective treatment modality, but much like the other published studies involving muscle energy technique, the researchers attribute these results to the small sample size of the study. The researchers ultimately ruled the study inconclusive, for a larger sample size is necessary to access the statistical significance of the treatment method on patients with COPD [19]. Although muscle energy techniques can be applied to any area of the body, unfortunately, they are very few publications of case studies investigating the effectiveness of muscle energy technique due to its recent emergence as a treatment modality [20]. However, nearly all the studies conducted to date show statistically significant results supporting the use of muscle energy treatment as an effective course of treatment [13,14,15,16,17,18,19]. In this respect, the few inconclusive studies are likely due to small sample sizes and, consequently, skewed data and insufficient evidence of its efficacy [19]. Hence, corroboration of the benefits and efficacy of OMT needs to be properly addressed via large multi-center clinical trials so these treatment approaches can be implemented regularly in the hospital setting. 

### 2.7. Benefits of Muscle Energy Technique

The advantages of muscle energy technique are immense. It is more affordable and provides easier access to healthcare, a decreased prevalence of the use of pain medication, and avoidance of expensive and unnecessary scans and procedures. The osteopathic approach takes into consideration the body as a single, functioning unit, approaching treatment from a mind, body, and spirit approach [13]. Although the research available at this time is limited, there is hope amongst the osteopathic community that this treatment modality, as well as the various others of osteopathic manipulative treatment, will become a new area of research and use within the medical field. Osteopathic medicine is emerging and becoming increasingly popular, with evidence of its success in clinical practice settings and in the emergency department of hospitals [49]. 

## 3. Myofascial Release Technique

Myofascial release (MFR) technique is a passive technique, either direct or indirect, that utilizes the fascia of the body to release binding tissues and stimulate the focus of the body’s processes on healing and health [21]. One of the main conditions utilizing MFR is chronic low back pain; however, other pathologies have also been treated, such as musculoskeletal, peripheral nerve, and tendon disorders. The MFR approach targets the myofascia, which is continuous with the connective tissue throughout the body and envelopes and supports bones, muscles, tendons, organs, vessels, nerves, and the lymphatic systems. The collagenous tissue is stretched across the entire body to create stability but is also elastic enough to ensure the pliability of the tissues [21,22,23]. Fascia can be thought of as one continuous sheet of tissue that connects the different parts of the body. A dysfunction somewhere in the fascia can be detrimental to other components of the internal body systems [22]. 

Identifying myofascial dysfunction requires a complete patient history and physical evaluation based on the presenting complaints [21,23]. The goal is to improve restriction, restore function, and decrease the patient’s discomfort [24]. Dysfunction is still identified based on the T.A.R.T. findings, as discussed earlier in this review. The body is evaluated on a holistic basis, examining how each component of the body connects to the next to determine the origin of the issue rather than just the presenting symptom [23]. Contraindications to MFR may include refusal by the patient, recent fractures, open wounds, deep venous thrombosis, and aortic aneurysm [21]. 

MFR requires engaging the fascia with gentle and consistent pressure either at the direct barrier or at the indirect position of ease, considering rotation, extension/flexion, and side bending of the tissue that is being treated [21,23]. The physician should be able to feel the tissue under his/her hands release and soften as the changes occur. Once the release is felt, the physician must take great care to slowly move the tissues back to normal rather than allowing an immediate return. Returning the tissues too quickly could cause a re-binding of the tissues, and the treatment would be futile. With the response of the tissues, results should be immediate. However, it may take a few days for the full treatment response to occur as the patient’s body is readjusting to the new fascial position and recovery [23,24]. 

There has been some research completed regarding the biomechanical and chemical effects of MFR, which provided evidence of decreased inflammation as well as an increased immune response [23]. At the cellular level, MFR treatment can alter the function of fibroblasts, which are specialized cells integral to fascia function within the body [50]. If the fibroblasts in the body are compromised, then healing and fascial function are also compromised, which leads to the diagnosis of somatic dysfunction. There is limited research in the effectiveness of MFR; however, the evidence-based research which has been completed shows significant positive results [51]. One identified study of MFR showed statistical significance in a randomized trial with improvement in both pain and disability compared to those who did not have myofascial release treatment [24]. Other studies have shown that MFR has been successful with temporomandibular joint (TMJ) disorders as it is related to the musculoskeletal system within the cranial region, where fascial tissues are able to be remobilized with an improvement of pathological barriers [51]. MFR, being a non-pharmaceutical and hands-on clinical technique, is an accessible treatment and recovery option for all patients, especially to decrease the need and use of pharmaceuticals in the population [21,23,24,51]. 

## 4. Balanced Ligamentous Tension Technique

Balanced ligamentous tension (BLT) is an indirect, passive technique used for ligamentous joints, such as the pelvis, innominate, shoulder, knee, elbows, etc. [25]. After a full patient history and physical evaluation, indications of BLT are based on a minimum of two T.A.R.T. findings, just as any other osteopathic technique requires. Contraindications of BLT are fractures, malignancies, and patient refusals. In addition, the skills of administering the BLT are dependent upon the palpation abilities of the physician to feel tissue changes [25]. 

Briefly, the BLT technique requires either a short or a long lever for the compression of joints [25]. The short lever applies compression across a shorter distance, while a long lever is applied across a larger distance. For example, a short lever for the shoulder may utilize the elbow, while a long lever would utilize the whole upper limb. Additionally, inducing rotation of a vertebra by contacting the spinous process is an example of a short lever, whereas inducing rotation of that vertebra by grasping an arm or leg and moving the body into rotation is an example of a long lever. Based on the restriction found, the distal hand monitors the joint while the proximal hand positions the lever. Compression should be added first, and then rotation, extension/flexion, and side bending should be utilized to place the joint towards its position of ease for an indirect treatment [25]. As the name suggests, a point of balanced tension is reached by fine-tuning the direction of pressure and movement induced with the provider’s hand. The point of finding the balanced tension in the ligamentous tissues is the key component that separates this technique from MFR. 

The ligamentous, or myofascial in the case of MFR, tissues are brought to a point of ease and directional equilibrium or balance, as the technique name suggests. The balanced tension is a point in which the joint feels as though it has equal tension in all directions from the ligaments that are attached to it. The feeling of release should be of warmth, unwinding of tissues, and increased motion to where less restriction is felt [25]. The process can be repeated after re-assessment is completed. When moving the joint back to neutral, the motions should be made slowly to ensure the joint does not go back into a state of restriction. Ultimately, however, being such a safe, tolerable treatment, BLT is one that can be utilized in many situations. One study utilized BLT in the case management of the whole body for an ACL injury, and another utilized BLT with TMJ involvement [25,26,51]. In both instances, osteopathic treatment aided in decreasing recovery time and improving both the function and healing capabilities of the body. 

## 5. Diaphragm Treatment Technique

Humans have five connective tissue structures that are arranged in a transverse pattern throughout the body, and these structures make up our diaphragms [27]. These five diaphragms include: the tentorium cerebelli, the tongue, upper thoracic diaphragm, respiratory diaphragm, and pelvic diaphragm. These structures play an integral role in our body to maintain homeostasis and proper function. Therefore, the diaphragms assist us in controlling and synchronizing our intracavity pressures. The diaphragms also assist in the control of circulation between the different cavities and the interstitial space of the visceral parenchyma [27]. 

The diaphragm is the primary muscle of respiration, so restrictions in this muscle can cause difficulty breathing. Treatment of the diaphragm should be considered in patients with COPD, emphysema, asthma, pneumonia, and other respiratory diseases that can be associated with shortness of breath in addition to general osteopathic treatment with the goal of supporting and improving homeostasis. Contraindications to diaphragm technique include patient refusal or intolerance, open wounds, and recent fractures. The respiratory diaphragm is the primary muscle involved in respiration, and it acts as a passageway for numerous vessels, nerves, and organs. The vena cava, esophagus, aorta, azygos root, lymphatics, and sympathetic nerves all pass through the respiratory diaphragm, so dysfunctions can potentially impact these important structures [28]. The respiratory diaphragm has attachments to the lower ribs, vertebra, and the xiphoid process, as well as numerous fascial attachments. These attachments are important as diaphragmatic dysfunctions can cause biomechanical problems elsewhere in the body as tension is transferred through the fascia. There are multiple approaches to treating the muscular diaphragm osteopathically. The following section discusses the diaphragm doming and indirect diaphragm release techniques alongside their uses.

### 5.1. Diaphragm Doming

One of the most common and effective approaches is called diaphragm doming, and this technique involves locating the xiphoid process and the costal arch on the anterior chest. Once located, the practitioner places their thumbs and thenar eminences approximately 2 to 3 inches below the costal margin (Figure 4), emphasizing the contact towards the undersurface of the diaphragm on expiration [29]. The patient will take deep breaths, and as the patient exhales, the practitioner will follow the diaphragm cephalad and exaggerate the motion with added pressure. The patient and physician will repeat this cycle three to five times, with the physician maintaining constant cephalad pressure during inhalation and increasing the pressure during exhalation.

Regarding the diaphragm and its related structures, this technique has been shown to help decrease prolonged pain that can occur in the cervical spine, as the diaphragm is innervated by the phrenic nerve (C3-C5). A study was published in 2016 in which the authors aimed to demonstrate the effect of treating distal tissues that are neurologically related to the originating spinal segments [29]. In this study, pain pressure thresholds were measured bilaterally in the C4 paraspinal musculature, the lateral end of the clavicle, and the upper third of the tibialis anterior before and after the diaphragm release treatment. After the treatment, the results showed a statistically significant hypoalgesia in the spinal segment of C4 bilaterally [30,31]. This result demonstrates that diaphragm treatment can induce an immediate effect on C4 due to its relation to the phrenic nerve. 

### 5.2. Indirect Diaphragm Release

Another frequently used treatment for the respiratory diaphragm is an indirect treatment which involves moving the tissues away from their restrictive barrier. The treatment requires the operator to place their hands on the antero-lateral surface of a supine patient and move the ribcage and overlying tissues into a position of ease by balancing it in the three planes of motion that exhibit the least amount of resistance. The patient will then take a series of deep breaths while this position is held by the operator, and the operator can make small changes as the diaphragm releases after each breath. A study by Mancini et al. (2019) [32] suggested that the indirect diaphragm release technique can improve diaphragmatic mobility by performing this technique as well as a treatment of the diaphragmatic pillars on healthy participants and then evaluating diaphragmatic motion and thickness using ultrasound assessments. The results of this study showed a statistically significant increase in diaphragmatic motion after the osteopathic treatments and therefore recommended further studies to be done to confirm the findings as well as identify clinical conditions that may benefit. 

The diaphragm is anchored to the thoraco-lumbar junction of the spine, the ribcage, and the psoas and quadratus lumborum muscles posteriorly. Anteriorly, the diaphragm is woven into the core muscles and lower ribs. Somatic dysfunctions in the respiratory diaphragm can translate to these other connections and cause pain that is relieved by treating the diaphragm. Martí-Salvador et al. (2018) [33] worked to show this relationship by using an OMT protocol involving diaphragm interventions on patients with non-specific chronic low back pain, and the results of this study showed that there was a statistically significant decrease in pain reporting in the experimental group as compared to the sham group [33]. This study demonstrated the benefit of diaphragmatic interventions in patients with non-specific chronic low back pain.

Each diaphragm plays an integral role in the overall function of the human body, and the goal of osteopathic medicine is to achieve balance in the human body so that it can function at its maximum potential. Although not individually discussed in this article, the other diaphragms in the body should also be considered during the above two treatments. Overall, the respiratory diaphragm has many functions in the human body, and being able to treat it with osteopathic manipulation has shown to be very beneficial in many circumstances. A variety of benefits can be derived from the multitude of connections that the respiratory diaphragm has in the body, from nervous system relationships to musculoskeletal and structural relationships. 

## 6. High-Velocity Low-Amplitude Technique

High-velocity, low-amplitude (HVLA) manual manipulation is a technique that is commonly used by many osteopathic physicians and chiropractors to treat pain or loss of motion in a joint. This modality carries more caution with it due to the nature of a rapid thrust being applied to the body. Contraindications may include rheumatoid arthritis, other inflammatory arthritis, Down syndrome, Chiari malformation, fractures, dislocations, joint instability, joint fusions, joint infection, myelopathy, bony malignancy, recent trauma, hypermobility, spondylolisthesis, and implanted devices. This technique involves a manipulation in which an operator provides a rapid (high velocity), therapeutic force that travels a short distance (low amplitude) within a range of motion in a joint, and this force will engage the restrictive barrier to release the restriction in the joint and restore range of motion [34]. One of the hypotheses that explain why HVLA is effective involves the concept that the thrusting motion stretches a contracted muscle, and this stretch can produce several afferent impulses from the muscle spindles that travel to the central nervous system. The muscle spindles are much more responsive to smaller amplitude stretch, and this makes them the ideal target for HVLA techniques [35]. The central nervous system will then send an inhibitory signal to the muscle spindle to relax that contracted muscle [34]. Therefore, the HVLA technique can be used on essentially any joint in the body, but this section will be focused on the uses and applications related to HVLA done on the cervical spine, thoracic spine, lumbar spine, and lumbosacral junction.

### 6.1. Thoracic and Lumbar HVLA

One of the most common ways to perform HVLA on the thoracic and lumbar spine is to have the patient lying in the lateral recumbent position on their non-dysfunctional side, with the operator standing by the side of the table facing the patient [36]. While monitoring the dysfunctional spinous processes, the physician flexes the patient’s top leg at the knee and hip until flexion is felt at the monitoring hand. The operator uses the forearms to induce an opposite rotation of the shoulder and pelvis. Each time the patient exhales, more rotational “slack” is taken up until an endpoint is felt. During the final exhale, when the patient relaxes, a final thrust is applied. There are other HVLA techniques that can be utilized on a patient if they are not receptive to this technique or if it is contraindicated. It is important to become familiar with the different variations of HVLA techniques used on the spine. 

HVLA has important benefits in increasing the range of motion (ROM) in a joint [37]. Griffiths et al. (2019) showed the immediate effects of HVLA on the thoracolumbar junction with regard to ROM [52]. This study involved a group of participants that received HVLA, a sham group that received light touch, and a control group that lay supine for the allotted time. After measuring ROM with an Acumar digital inclinometer before and after treatment, there were significant increases in ROM noted in the group of participants that received HVLA treatment [53,54]. Another notable study with relevance to patient care utilizing HVLA was published in 2013, which compared the effectiveness of HVLA with diclofenac, a nonsteroidal anti-inflammatory drug [36]. This 2013 study by von Heymann et al. was done on patients with acute low back pain for less than 48 h, and the results showed that the group of participants that received the HVLA manipulation had a significantly improved Roland–Morris Disability Score. Although this study was unable to use a control group to strengthen the study for ethical reasons due to unsustainable pain, the results show promise in using HVLA to relieve acute low back pain from structural dysfunctions as a superior alternative to diclofenac.

However, it is also noted that some of the absolute contraindications to be aware of before HVLA treatment on the thoracic/lumbar spine are bone compromise (trauma, tumor, infection), neurological issues (spinal cord compression, cauda equina syndrome, etc.), vascular compromise (vertebrobasilar insufficiency, cervical artery abnormalities, aortic aneurysm, acute abdominal pain), and increased risk of harm to the patient (lack of skill, lack of consent, or lack of diagnosis) [34,35,36].

### 6.2. Cervical HVLA

The procedure of cervical HVLA will now be discussed as we reference studies and applications of this technique. The process of cervical HVLA begins with obtaining a diagnosis of the somatic dysfunction, then ruling out existing contraindications that the patient may have. HVLA techniques will be optimally successful if the patient is fully relaxed, so performing soft tissue massage and/or myofascial release on the area before treatment can help to build trust and relax the patient [37]. For the treatment, the patients lay supine with the operator sitting at the head of the table. While supporting the head and monitoring the cervical spine at the level of the dysfunction, the operator will engage the restricted barrier in the three planes of motion. Once the patient relaxes, they will begin taking deep breaths. During each exhale, the operator will engage the barrier more until an endpoint is felt. On the final exhale, a short thrust is applied to move the segment through that final restrictive barrier. Cervical HVLA can also be done in a seated position utilizing similar conceptual and positional movements as the supine treatment position to achieve the same result. However, the supine position is more commonly utilized. 

Applying HVLA to the cervical spine has been shown to exhibit many benefits as well as contraindications (Table 2). Many of these benefits are listed, along with relevant absolute contraindications to the technique. Cervical pain is associated with disability and significant health costs as well as being classified as one of the top two reasons for disability caused by musculoskeletal pain conditions by the Global Burden of Disease studies [54]. It has been shown that cervical HVLA may be effective in resolving pain in the neck, shoulder, and head (including cervicogenic headaches) as well [37]. A notable benefit of HVLA to the cervical region involves the manipulation of the atlanto-occipital joint, resulting in the patient experiencing an immediate increase in the pain-pressure threshold of trigger points on the masseter muscle and the temporal muscle. This atlanto-occipital manipulation also showed an increase in the degree of active mouth opening [54]. Overall, HVLA has been shown to be an effective tool used by osteopathic physicians, chiropractors, and physiotherapists to treat pain and increase ROM in patients. As more thorough studies are completed on this technique, there will be more insight as to the exact mechanisms of benefit from this treatment. With informed consent, proper preparation of the patient, confirmed lack of contraindications, and correct technique, this treatment has been shown to decrease pain, increase ROM, and improve the quality of life for patients with joint immobility.

## 7. Rib Raising Technique

Rib raising is an osteopathic technique that can be used in many patients, especially those with disorders involving the sympathetic nerves [42]. Contraindications to rib raising may include recent fractures, patient refusal or lack of consent, bony malignancy or infection, and any relative considerations given the clinical scenario. The rib raising technique focuses on the sympathetic chain ganglia that run parallel with the thoracic spine, and the stimulation of this chain is thought to help to bring the autonomic nervous system into a balanced state (Figure 5). 

The rib raising technique is performed with the patient in a supine or seated position, and the physician sits on the side of the patient with their fingertips contacting the ipsilateral rib angles (Figure 6). The physician will use a slow and gentle rocking motion to lift the rib angles anteriorly and then release the tension [43]. The physician will work their way along the thorax until the technique has been done on each rib angle multiple times, ensuring this technique is also done bilaterally [43,44]. Indeed, there have been a variety of modifications made to this technique [45,55]. The physician will typically feel the muscle tension and motion restrictions decrease as the technique is performed on the patient. Rocha et al. (2020) [55] carried out a pilot study to measure the effects of rib mobilization and diaphragm release on cardiac autonomic control in patients with COPD. It was documented that the majority of patients reported significant symptom improvement. 

In a 2010 study, Henderson et al. monitored biomarkers such as salivary α-amylase to measure the autonomic nervous system effects of rib raising [56]. Salivary α-amylase is known to increase during states of physical or psychological stress in humans, so this study involved measuring the levels of α-amylase in participants before, immediately after treatment, and 10 min after treatment. The study took note that the salivary flow rate, which is mostly dependent on parasympathetic stimulation, did not show any significant changes with the technique. The study did show that there was a significant decrease in salivary a-amylase in the treatment group that received rib raising compared to the placebo group that received light touch only [56]. This study’s results support the idea that rib raising can decrease sympathetic nervous system activity, which can prove beneficial in numerous conditions. 

The reduction in sympathetic activity from rib raising can be beneficial in patients suffering from pneumonia. The technique can augment lymphatic flow by increasing respiratory excursion. Excessive sympathetic stimulation can reduce chest wall mobility by generating hypertonicity of the rib cage musculature and increasing intra-abdominal pressure, and lymphatic flow depends on this pressure gradient to flow properly [57]. Rib raising can decrease sympathetic activity, and this causes increased chest wall mobility and improved lymph flow. Therefore, this technique could be used alone or with other OMT techniques to benefit patients with pneumonia. In general, it has been frequently observed that the use of osteopathic techniques in a hospital setting is valuable, as many patients cannot ambulate properly, which limits the patient’s ability to receive other beneficial treatments, such as physical therapy. In these patients, osteopathic techniques can be used to treat restrictions directly or indirectly, and they can be modified to allow effective treatment for patients who cannot actively participate in the treatment. Rib raising is a great example of this passive, but effective, treatment technique. 

In a recent study, conducted on eighty-seven hospitalized, non-intensive care unit patients to determine the tolerance of rib raising in this population, Chin et al. (2019) clearly documented that rib raising is well tolerated [44]. They treated the patients with rib raising and then requested that they rate the treatment on a scale of 0 to 10, where 0 is no discomfort at all and 10 is maximum discomfort. The results showed that the treatment is well tolerated, with 92.0% of patients scoring between 0 and 3, 6.9% scoring between 4 and 6, and 1.1% scoring between 7 and 10 on the tolerance scale [44]. Rib raising can also be used in coordination with many other osteopathic techniques, and it can amplify the effects of other treatments. Interestingly, Martingano et al. (2019) utilized an array of treatments such as rib raising, lymphatic drainage, suboccipital decompression, paraspinal inhibition, and sacral inhibition as part of labor management for pregnant patients to determine if OMT had any beneficial effects. The group that received the OMT combination had a significantly shorter labor duration than the control group [58].

## 8. Lymphatic Pumping Technique

The lymphatic pumping technique (LPT) has been the most well-studied of the OMTs and has shown significant promise in enhancing the immune defense against microbial infections. Indeed, the lymphatic system is a vital component of homeostasis and immune responses in the body [38]. Briefly, the lymphatic system involves lymphatic vessels, capillaries, and lymph nodes/organs that work together to absorb interstitial fluid, transport lymph, and accelerate immune function. When lymphatic capillaries absorb excess interstitial fluid for transport, they are also transporting osmotically active proteins, parenchymal cell products, inflammatory mediators, immune cells, proteins, apoptotic cells, antigens, and infectious organisms through to the lymph nodes for processing. Lymphatic fluid is transported through the vessels via extrinsic and intrinsic forces. At rest, approximately 1/3 of lymphatic fluid transport in the lower extremities is due to compression of lymphatic vessels by skeletal muscle contractions (extrinsic), and 2/3 is from active pumping of the smooth muscle in the lymphatic vessels (intrinsic) [39]. The lymphatic fluid moves through the body at approximately 125 mL/h at rest, and it can increase by a factor of 10 during physical activity [40]. There are numerous problems that can occur with this system that can disrupt the homeostasis of the body, such as: valve failure (valves normally prevent the backflow of lymph), fluid overload, inability to contract muscles, damage to lymphatic vessels/nodes from surgery or radiation, etc. When dysfunction of the lymphatic system occurs, it can lead to edema, inflammatory mediator accumulation, tissue injury, poor immune system function, and a variety of other disease states [38]. Contraindications to lymphatic treatment may include anuresis and/or kidney failure, advanced heart failure, acute asthma, unstable cardiac conditions, and acute fractures in the area of treatment. 

The importance of the lymphatic system has been a major focus of osteopathic physicians from the beginning, and many techniques have been designed to improve lymphatic circulation [41]. Indeed, the LPT has been used by osteopathic physicians in the management of congestive heart failure, upper and lower gastrointestinal tract dysfunctions, respiratory diseases, infection, and edema [52]. There are LPTs that are used for different regions of the body, and osteopathic physicians frequently use a combination of techniques to achieve optimal lymphatic circulation. Rand et al. (2015) [59] showed positive evidence of integrating osteopathic medicine into primary care using the ‘pedal pump’ and the ‘thoracic pump’ approaches to treat patients. The pedal pump involves standing at the feet of a supine patient and gently applying dorsiflexion (Figure 7A). The gentle force of dorsiflexion causes a fluid wave to move cephalad within the patient, and then the fluid wave will rebound and move caudally. The physician maintains this oscillation of fluid within the patient to promote whole-body fluid circulation within the patient. The thoracic pump is another common technique with a similar principle. The patient is placed supine as the physician stands at the head of the bed with their hands on the patient’s thoracic wall (Figure 7B). A gentle rhythmic pumping motion is initiated that promotes lymphatic movement within the thoracic ducts and the cisterna chyli. In addition to the ‘pedal pump’ and ‘thoracic pump’ techniques, there are numerous other LPT techniques that focus on releasing tension in the diaphragms in order to decrease restrictions of lymphatic flow that have shown promising effects in both human and animal models.

There have been studies done on rodents and dogs to show that there is an increase in lymphatic flow as well as increases in leukocyte concentrations in the lymphatic fluid during and after lymphatic pump treatments [38]. In a 2013 study involving lymphatic duct samples taken from dogs before, during, and after two rounds of LPT, there were significant increases in leukocyte numbers within 1 min of the treatment beginning. This suggested that the lymphatic pump assisted in the mobilization of the leukocytes into circulation. These numbers remained elevated at 10 min post-treatment. This increase in immune-support cells within the lymphatic fluid could benefit the health of patients who are unable to effectively circulate their own lymph [38].

One of the most important implications of the LPT is its efficacy against microbial infections. Studies have documented that lymphatic pumping is an effective procedure in clearing bacteria, particularly in pneumonia patients. A study that supported this claim was done by inoculating rats with *Streptococcus pneumoniae* and measuring the total number of bacteria and leukocytes found in the lungs 8 days post-inoculation [60]. They compared a group of rats that were treated with a daily lymphatic pump technique to a sham group and a control group. The sham group received light touch daily, and no compressions were done. After the 8 days, the group of rats which received treatment had significantly fewer bacteria within their lungs than the sham or control groups. The treated rats also had decreased leukocyte numbers in their lungs. It was proposed that the lymphatic treatment assisted in mobilizing the immune cells by rapidly transporting antigens to the lymph nodes, which assisted the immune system activation in fighting off the bacteria [60].

LPTs also have relevance to the impact of vaccines. A study was completed in 1998 in which subjects were given the hepatitis B vaccine, and one group received OMT in the form of lymphatic and splenic pumps while the other group received light touch only [61]. The results of this study showed that from the 6th week post-vaccination, the treatment group had higher average anti-hepatitis B titers than the control group. This research supports the notion that this lymphatic pump technique can boost the immune system response and could be a useful adjunct to enhancing vaccine responses in the future. Indeed, several recent reviews have emphasized the advantage of using OMT as an adjunct to current allopathic treatments against infections and emerging pandemics [62,63,64]. In this respect, promising evidence regarding the beneficial effects of LPT in patients with progressing COVID-19 pneumonia has initiated several multicenter clinical trials [65]. Therefore, it is imperative that more research is conducted in the future to corroborate these findings and elucidate the molecular mechanisms linked to the significant beneficial effects. 

## 9. Cranial Osteopathy

Cranial osteopathy is widely known for the treatment of babies but is equally effective for children, adults and the elderly. This OMM technique involves different treatment modalities that improve the primary respiratory mechanism via balancing cerebrospinal fluid (CSF) flow, somatic dysfunction, parasympathetic regulation, and the inherent motility of the CNS [46,47]. Cranial osteopathy is a subtle and refined approach to osteopathy that follows all the principles of osteopathy and takes into account both the anatomy and physiology of the head. There are several cranial positional holds that can affect parasympathetic output, such as the vagus nerve impacting other connected processes that can cause somatic dysfunction. The motion within these cranial holds can help the practitioner determine the pathology of the physiological strain pattern and properly treat the problem. Importantly, the sacrum moves involuntarily between the ilia due to its dural attachments through the spinal column to the cranium. The dura maintains the structural integrity of the bony cranium; hence, if one end is affected, the problem is transmitted throughout. Osteopathic physicians are well trained in providing balance to the entire system through cranial OMT. Contraindications to cranial technique include patient refusal, intracranial bleeding, and skull fracture, plus any other relative considerations given the clinical scenario.

Cranial techniques can also be implemented in treating residual concussion symptoms. Cases have been reported that show cranial techniques after a concussion helped patients return to their daily activities [66]. While up 90% of patients will have a resolution of concussion symptoms, the remaining 10% will have to experience symptoms for an extended period of time. The benefit of osteopathic cranial medicine is not just supported by anecdotal evidence. Alzheimer’s disease is a debilitating and devastating disease to not only the patient but their support system. Cranial manipulative medicine has been shown in an aged rat model to augment the effects of aging and represent Alzheimer’s disease [67]. This was shown by performing OMM on a rat for 7 days and then performing PET scans, learning and memory assays, and immunohistochemical studies. Western blot studies showed an increase in glial fibrillary acidic protein (GFAP), aquaporin 4 (AQP4) and lymphatic vessel endothelial hyaluronan receptor 1 (LYVE1). Compared to the control, the cranial manipulation group showed improved spatial memory and increased fluid circulation, which improved the removal of metabolic waste. The interest in nontraditional medicine and alternative techniques for symptom relief is not a new interest. Dating back to the 20th century, Abraham Flexner wrote about the addition of osteopathic medicine to scientific education and research [48]. Flexner’s thoughts about integrating additional medical techniques into education have been influential to modern medical teachings and become part of the curricula and values taught by medical education.

## 10. Conclusions

Osteopathic manipulative treatment encompasses a variety of techniques targeted at enhancing the body and its homeostatic function as a holistic unit. It is incredible to see the capability the human body has for healing itself, given the proper conditions. As previously mentioned, osteopathic medicine is more affordable and far less invasive than other medical practices. It also promotes full body healing, as it considers the body as a single, functioning unit. However, even with the promise of harnessing the body’s tremendous healing capacity, the use of OMT can be severely limited due to a variety of factors. First and foremost is consent. Whether or not a patient either is willing or wishing to be treated with OMT or not dictates the utilization. The second is the clinical scenario and prioritizing of the immediate, urgent, and emergent needs of the patient. Third, many patients hang in the balance of a dysfunctional state that may be susceptible to further decompensation if they are treated too aggressively with OMT. In light of these limitations, it is advisable to use caution and ‘do no harm first’.

While the research determining the successfulness of OMT on various symptomology is limited, there is new evidence to emerge in the coming years as its popularity increases along with the awareness of this field of medicine. When osteopathic physicians incorporate OMT into every patient encounter as part of the physical exam, it allows the physician to get a full-body picture and better understand the history of the patient and the details of the current illness/symptom(s) the patient may be facing. As research, techniques, and interest in osteopathic treatments grow, the data will continue to show its efficacy and safety. Modern healthcare has shifted to evidence-based medicine, which benefits the care of patients, and this review helps summarize the current knowledge of OMT. With limited contraindications and many indications, the use of OMT for various disease states will increase. OMT continues to grow and flourish in the medical field, and it is the hope of osteopaths for it to become a common component of physical exams and medical management across all healthcare fields due to its many benefits and successful outcomes. 

## Figures and Tables

**Figure 1 medicines-09-00033-f001:**
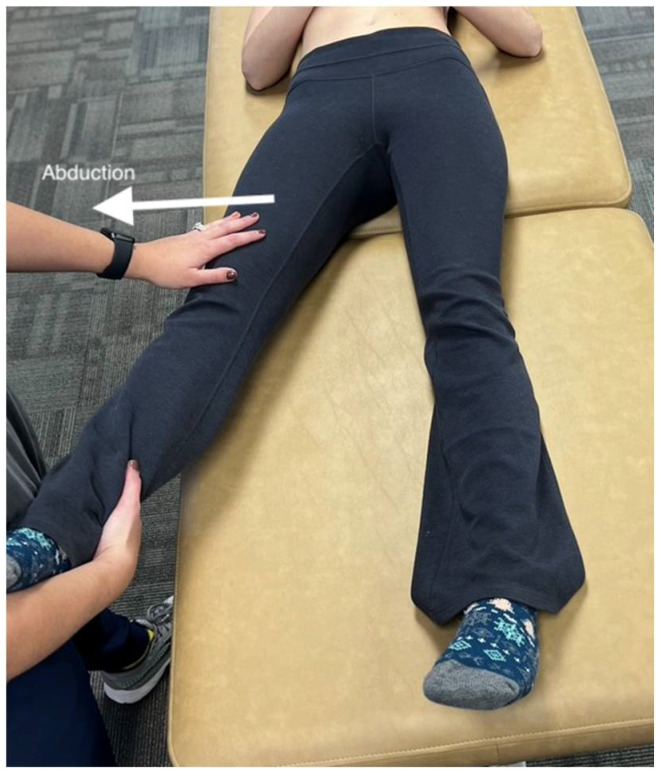
The principles of palpatory diagnosis and manipulative technique. The figure shows the procedure for the abduction of the legs. The osteopathic physician compares abduction of both left and right legs to determine which muscle to treat. Whichever leg has greater restriction compared to the other would be the leg selected for treatment.

**Figure 2 medicines-09-00033-f002:**
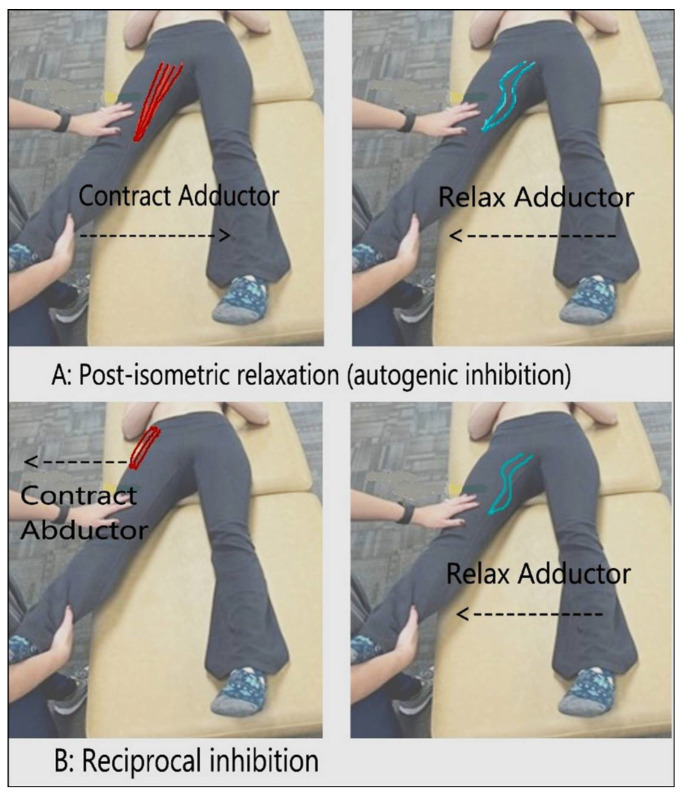
Post-isometric relaxation and reciprocal inhibition. Neuromuscular actions are mediated by Golgi tendon organs and inhibitory interneurons of the spinal cord [10]. (**A**) Contraction of agonist (adductor), and the reflexive post-isometric relaxation of the same muscle. (**B**) Contraction of antagonist (abductor), and the reciprocal inhibition/relaxation of the agonist (adductor).

**Figure 3 medicines-09-00033-f003:**
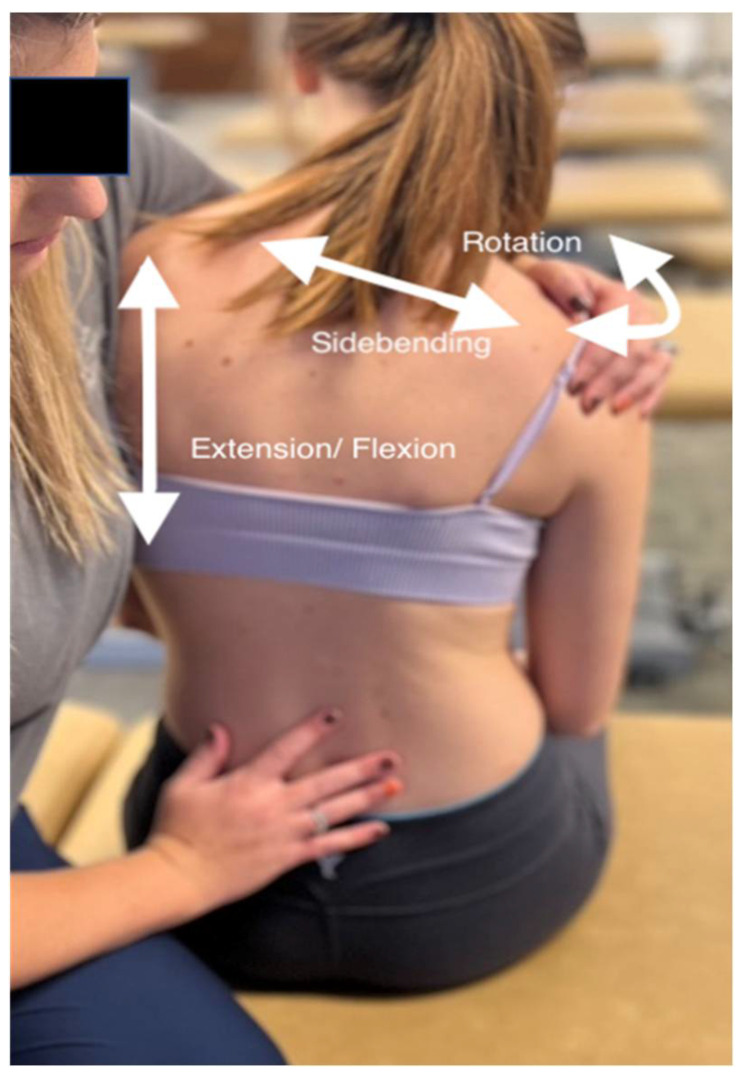
Muscle energy technique on the lumbar spine. The physician is in contact with the shoulders to control the position of the trunk and spine in 3 dimensions. The physician uses a free hand to monitor the lumbar spine for localization of the barrier at the appropriate segmental level. The monitoring hand will remain on the vertebral processes of the somatic dysfunction to detect the tissue’s responses to vectors.

**Figure 4 medicines-09-00033-f004:**
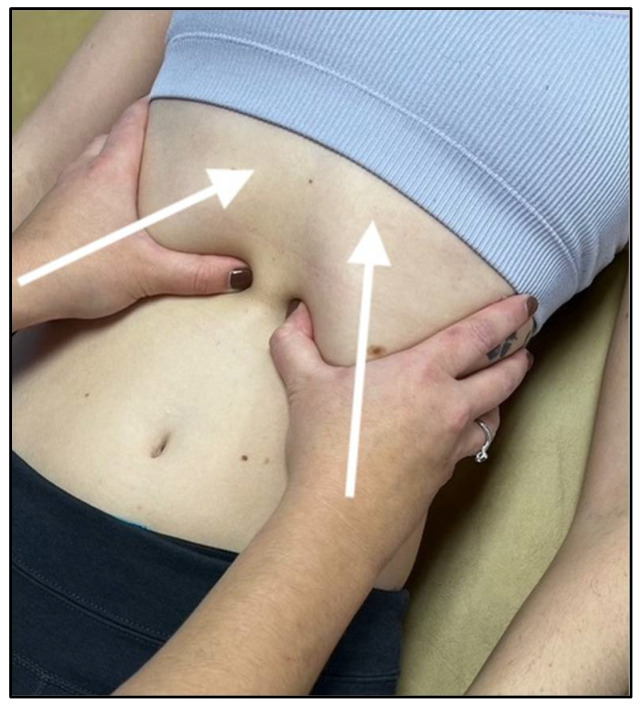
Hand placement and force vectors for diaphragm doming technique.

**Figure 5 medicines-09-00033-f005:**
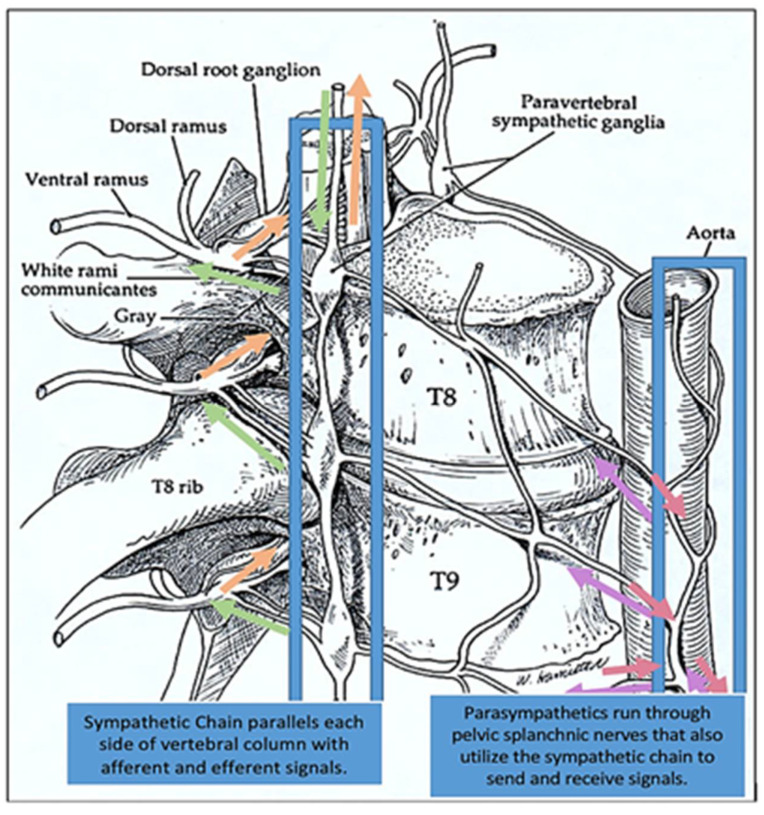
Sympathetic chain ganglionic pathway. The bilaterally symmetric sympathetic chain ganglia are located just ventral and lateral to the spinal cord and run parallel with the thoracic spine. The sympathetic chain ganglionic pathway extends from the upper neck down to the coccyx, forming the cervical, thoracic, lumbar, or sacral ganglions. The postganglionic fiber extends to the thoracic cavity, abdominal cavity, or pelvic cavity, and dysfunctions in many organs can be addressed by different variations of the rib raising techniques. Adapted from Karemaker et al. (2017) [45]. The eighth and the ninth thoracic vertebrae are designated as T8 and T9, respectively. The eighth and ninth thoracic spinal nerves run beneath these vertebrae. The green arrows designate afferent neurons and the brown arrows designate the efferent neurons.

**Figure 6 medicines-09-00033-f006:**
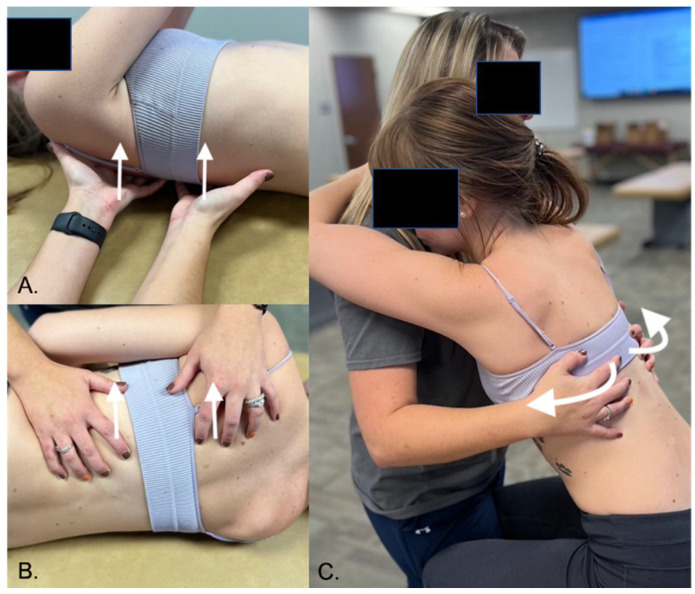
Rib raising technique variations. (**A**) The osteopathic physician has the patient in a supine position. The hands are positioned at the position of the rib angles to gain proximity to the thoracic chain ganglia. With the fingers placed at the rib angles, pressure can be applied through the shoulders and the elbows into the wrists. This position is held until the soft tissues release. Once soft tissue release is appreciated, the hands are then repositioned to subsequent ribs. (**B**) Lateral recumbent positioning. (**C**) Seated positioning.

**Figure 7 medicines-09-00033-f007:**
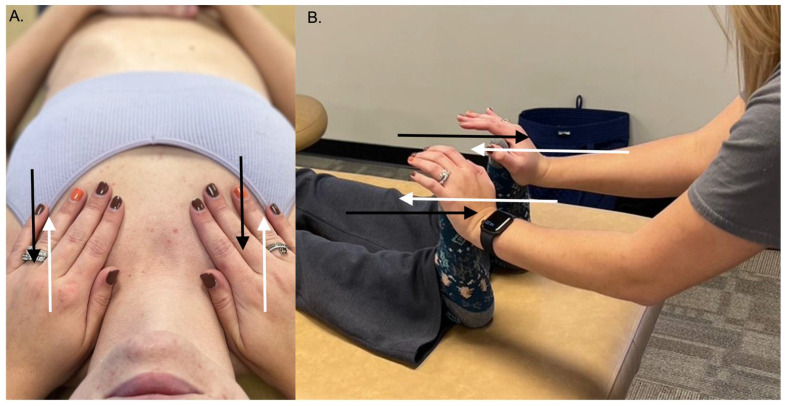
Lymphatic pumping techniques. Both the thoracic (**A**) and pedal (**B**) pump techniques can be used to enhance the body’s lymphatic flow. The white arrows in each panel indicate the physician’s force inducing fluid movement, while the black arrows indicate dynamic fluid movement back towards the physician to create a continuous, rhythmic oscillating motion.

**Table 1 medicines-09-00033-t001:** Osteopathic manipulative techniques: their descriptions and uses.

Techniques	Descriptions	Uses	References
**Muscle Energy**	Post-isometric relaxation—relaxing and lengthening a hypertonic muscle via engagement of the agonist muscle group. Reciprocal inhibition—relaxing and lengthening a muscle by activating the stretch reflex of muscle spindle fibers of the antagonist muscle, causing the agonist muscle to reflexively relax.	Increase range of motion of restricted areas, stretch tight muscles, reduce chronic pain, and improve circulation and lymphatic flow throughout the body.	[7,8,9,10,11,12,13,14,15,16,17,18,19,20]
**Myofascial Release**	Indirect or direct techniques: Use of directionality and a passive approach by following the fascia in all directions of ease.	Release constricted tissues within the musculoskeletal systems to facilitate blood flow and lessen pain.	[21,22,23,24]
**Balanced Ligamentous Tension**	Techniques employ both compression and passive approaches to place a joint in “balance” when moved in different planes.	Increase range of motion in restricted joints in the whole body, such as the knee, TMJ, ankle, shoulder, fingers, etc.	[25,26,27]
**Diaphragm Doming**	Relaxing the respiratory diaphragm by applying pressure beneath the rib cage bilaterally.	Improve diaphragmatic excursion. Secondary uses: decrease cervical, thoracic, and lumbar pain due to multiple attachments and improve circulation.	[28,29,30]
**Indirect Diaphragm Release**	Relaxing the respiratory diaphragm by placing the hands on the antero-lateral rib cage and moving the tissues into their position of ease.	Decrease cervical, thoracic, and lumbar pain due to multiple attachments and improve circulation.	[31,32,33]
**High Velocity Low Amplitude (HVLA)**	Application of a rapid force over a short distance directed at a joint, which engages the restrictive barrier and releases the restriction.	Decrease joint pain, improve mobility, and improve range of motion.	[34,35,36,37]
**Lymphatic Pump**	Clearing obstructions to lymphatic channels and employing pumping techniques that are commonly performed at the feet, abdomen, and thorax.	Used in gastrointestinal tract infections, respiratory infections, and edema. *Secondary uses*: Improve immune function and vaccination efficacy.	[38,39,40,41]
**Rib Raising**	A method of ‘raising’ the ribs anteriorly with the intention of influencing the functin of the sympathetic chain ganglia that are anterior to the ribs.	Decrease sympathetic nervous system activity, increase respiratory excursion, increase chest wall mobility and lymphatic flow.	[42,43,44,45]
**Cranial osteopathy**	Improving both central and peripheral brain functions by balancing the CSF flow, and improving the motion of the sacrum and cranium.	Promote functioning of the primary respiratory mechanism, treat somatic dysfunction of the skull bones, tissues, dura, and overlying fascia, reducing symptom burdens related to concussions.	[46,47,48]

CSF: Cerebrospinal fluid.

**Table 2 medicines-09-00033-t002:** Cervical HVLA benefits and contraindications.

Benefits	Absolute Contraindications [42]	
Improved Range of Motion	Acute Fractures	Ligament Rupture
Improved Quality of Life	Soft Tissue Injury	Osteoporosis
Decreased Neck Pain	Acute Myelopathy	Patient Refusal
Decreased Shoulder Pain	Ankylosing Spondylitis	Recent Surgery
	Chiari Malformation	Rheumatoid Arthritis
	Connective Tissue Disease	Joint Fusion
	Dislocation	Tumor/Malignancy
	Down Syndrome	Vertebral Artery Abnormalities
	Infection	Vascular Disease
	Instability	

## Data Availability

Not applicable.

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
