# Peer review of "Osteopathic Manipulative Medicine: A Brief Review of the Hands-On Treatment Approaches and Their Therapeutic Uses"

_medicines, 2022, doi:10.3390/medicines9050033_

Round 1

Reviewer 1 Report

nice overview

Author Response

We thank the reviewer for the highly positive comments.  No criticism needed to be addressed.

Reviewer 2 Report

In this manuscript, the authors summarized current studies about Osteopathic manipulative treatments. Overall, this manuscript is a balanced review of relevant literature.

  1. In Page 2 Line 80, the subtitle should be concise.
  2. The language should be carefully checked throughout the manuscript. For instance, the possessive case should be avoided as they are not generally suitable in formal writing.
  3. In the figure caption of Figure 6, each image in this figure needs to be labeled and described.

Author Response

Reviewer 2:

We sincerely thank the reviewer for stating that this manuscript is a balanced review of relevant literature.

Critiques:

  1. In Page 2 Line 80, the subtitle should be concise.
  2. The language should be carefully checked throughout the manuscript. For instance, the possessive case should be avoided as they are not generally suitable in formal writing.
  3. In the figure caption of Figure 6, each image in this figure needs to be labeled and described.

Response:

  1. We have shortened the subtitle. This now appears on pg-2, line-86.
  2. We have carefully checked and corrected the language in the entire manuscript. The possessive cases have been removed.
  3. Each image on Figure 6 has now been labeled and described.

Reviewer 3 Report

Thank you for the opportunity to review the work. The work contains an interesting description of the methods used in osteopathic medicine. From the physiotherapist's point of view, the authors describe the methods very clearly, supporting the descriptions with references to the literature.

The article is an interesting study that describes Osteopathic Manipulative Medicine.
The goals of osteopathic medicine and osteopathic techniques have been clearly described. There are figures in the article showing the performance of the techniques mentioned in the text. An interesting supplement to the descriptions of techniques is the literature describing the effectiveness of using these techniques in patients.
 The article is easy to read and contains information that is not always widely available in other articles on osteopathic medicine. Below are some comments to the authors:
Please shorten the title of subchapter 1.1. Transfer the justification for the differences between DO and MD to the text constituting the content of the subchapter.
Short description of all Figures.
At the beginning of each chapter, please describe the indications and contraindications for using the described technique.
Remove the bold description from subchapters 6.1 and 6.2.
Please summarize the limitations of using osteopathic techniques.

Author Response

Reviewer 3

We thank the reviewer for stating that the work contains an interesting description of the methods used in osteopathic medicine, and that the description of the methods are very clear and is easy to read.

Critiques:

Please shorten the title of subchapter 1.1. Transfer the justification for the differences between DO and MD to the text constituting the content of the subchapter.

Short description of all Figures.

At the beginning of each chapter, please describe the indications and contraindications for using the described technique.

Remove the bold description from subchapters 6.1 and 6.2.

Please summarize the limitations of using osteopathic techniques.

Response:

- We have shortened the title of subchapter 1.1.

- We have transferred the justification for the differences between DO and MD to the later section.

- We have added a short description of all Figures.

- We have added the indications and contraindications under each OMT technique. 

- We have removed the bold description for the subchapters 6.1 and 6.2.

- We have included some of the current limitations (highlighted)..